# Type 2 Diabetes Mellitus Mediation by the Disruptive Activity of Environmental Toxicants on Sex Hormone Receptors: In Silico Evaluation

**DOI:** 10.3390/toxics9100255

**Published:** 2021-10-08

**Authors:** Franklyn Nonso Iheagwam, Joseph Kelechi Odiba, Olawumi Toyin Iheagwam, Olubanke Olujoke Ogunlana, Shalom Nwodo Chinedu

**Affiliations:** 1Department of Biochemistry and Covenant University Public Health and Wellness Research Cluster (CUPHWERC), Covenant University, Ota PMB 1023, Nigeria; banke.ogunlana@covenantuniversity.edu.ng (O.O.O.); shalom.chinedu@covenantuniversity.edu.ng (S.N.C.); 2Aafud Industry Nigeria Co., Ltd., Lagos 100212, Nigeria; kelechijoseph@gmail.com; 3Beloxxi Industries Ltd., Agbara 112102, Nigeria; wuminaira@gmail.com

**Keywords:** in silico, disruptive activity, environmental toxicants, type 2 diabetes mellitus, molecular docking, molecular dynamics, gene network analysis

## Abstract

This study investigates the disruptive activity of environmental toxicants on sex hormone receptors mediating type 2 diabetes mellitus (T2DM). Toxicokinetics, gene target prediction, molecular docking, molecular dynamics, and gene network analysis were applied in silico techniques. From the results, permethrin, perfluorooctanoic acid, dichlorodiphenyltrichloroethane, O-phenylphenol, bisphenol A, and diethylstilbestrol were the active toxic compounds that could modulate androgen (AR) and estrogen-α and –β receptors (ER) to induce T2DM. Early growth response 1 (EGR1), estrogen receptor 1 (ESR1), and tumour protein 63 (TP63) were the major transcription factors, while mitogen-activated protein kinases (MAPK) and cyclin-dependent kinases (CDK) were the major kinases upregulated by these toxicants via interactions with intermediary proteins such as PTEN, AKT1, NfKβ1, SMAD3 and others in the gene network analysis to mediate T2DM. These toxicants pose a major challenge to public health; hence, monitoring their manufacture, use, and disposal should be enforced. This would ensure reduced interaction between people and these toxic chemicals, thereby reducing the incidence and prevalence of T2DM.

## 1. Introduction

Endocrine-disrupting compounds (EDC) are natural and synthetic environmental toxicants that alter the endocrine system through numerous mechanisms, inducing adverse health effects in the exposed individuals and populations [1]. The introduction of these structurally diverse chemicals led to their increased use as well as domestic, occupational, and environmental exposure by humans. They are classified as different groups such as phthalates, polychlorinated biphenyls, polyaromatic hydrocarbons, dioxins, and other major groups distributed in different pesticides, chemicals used in industries, plastics, components of packaging, fuel, and so on [2,3]. Due to the reported numerous applications of over 500 of these compounds daily, they were implicated in the pathogenesis of numerous health effects. There is a strong cause and effect correlation between endocrine disruption, obesity, and type 2 diabetes mellitus (T2DM). Insulin resistance, hyperinsulinemia, dyslipidaemia, and hyperglycaemia are characteristic symptoms of impaired glucose and lipid homeostasis known to develop because of exposure to these compounds [4].

T2DM is mainly characterised by disrupted glucose homeostasis, with genetic and environmental factors implicated in its aetiology in addition to insulin resistance [5]. Recently, sex hormones (such as androgens and estrogens) and their receptors in peripheral tissues were reported to play an important role in gene regulation, insulin resistance, and glucose homeostasis [6,7]. Environmental toxicants, which are analogues of these hormones, bind to these receptors, alter their structure, and function concomitantly disrupting glucose homeostasis, insulin release, and action leading to the occurrence of T2DM [8,9]. Genomic studies corroborated the environmental, genetic, and epigenetic processes of T2DM, which steadily evolves during the lifetime of a patient. Nonetheless, accurate prediction for personalised medicine using genomic information is a challenge as drug discovery is driven by a complex interaction of proteins and cells working with environmental factors rather than genetic code. Also, genome-wide mRNA assesses cellular conditions under different states; they do not provide information on downstream mechanisms [10]. There is a paucity of information on the diabetic downstream effect of sex hormone receptor binding by EDC and cell signalling networks at a molecular level. This study aims to investigate the sex hormone receptors that are targeted and disrupted by environmental toxicants to mediate type 2 diabetes mellitus. It also identifies the downstream metabolic signatures that are upregulated upon disruption by the toxicants.

## 2. Methods

### 2.1. Toxic Ligand Preparation

A list of 19 environmental toxicants was identified from literature [4,11]. They were retrieved from PubChem Compound Database in their canonical simplified molecular input line entry specification (SMILES) format as shown in Table 1.

### 2.2. Toxicokinetic and Sex Hormone Receptors Target Prediction

The absorption, distribution, metabolism, and excretion (ADME) prediction of the toxicants were carried out using SwissADME server [12,13]. The identification of the toxicants as potential sex hormone receptor targets was predicted using the SwissTargetPrediction server with *Homo sapiens* selected as the target organism [14]. The toxicants’ SMILES format was inputted into both servers for toxicokinetic and target prediction.

### 2.3. Protein Preparation, Molecular Docking, Bioactivity, and Molecular Dynamics

Androgen (AR) and estrogen-α and -β receptor (ER) protein structures were downloaded from protein data bank (www.rcsb.org, accessed on 11 February 2021), with 1T5Z, 5KRJ, and 5TOA as the respective identification codes. The protein structures’ energy level without their complexed ligands was minimised by 3D refine server [15]. The toxicants’ SMILES format was converted to 3D protein data bank (pdb) format using Open Babel [16]. AutoDock Tools 4.2 was used to prepare ligands, remove water and heteroatoms from protein structures, add Gasteiger charges, and set grid box coordinates and size parameters for molecular docking, as shown in Table 2 at 1 Å space while molecular docking was carried out with AutoDock Vina, as previously reported [17,18,19]. The interactions between high-ranking toxicants (based on their binding score) and their respective receptors were visualised using BIOVIA Discovery Studio Visualizer, 2017 [20].

Metrics of binding effectiveness and toxicant bioactivities such as ligand efficiency [21], binding efficiency index [22], and lipophilic efficiency [23] were calculated as shown in Equations (1)–(3). The original ligand of AR, ER-α, and ER-β was deleted and redocked in the active site for accuracy and reliability verification of the molecular docking protocol used in Table 2.
(1)LE=−BE ÷HA 
(2)$$BEI=\frac{pIC_{50}}{MW\;\left(kDa\right)}$$
(3)$$LipE=pIC_{50}-logP$$

*LE*: ligand efficiency, *BEI*: binding efficiency index, *LipE*: lipophilic efficiency, *BE*: binding energy, *HA*: heavy atoms, *MW*: molecular weight, *pIC*_50_: predicted *IC*_50_. The binding energies were converted to *pIC*_50_ which can be interchanged with (K_i_) and dissociation constant (K_d_)_._

Molecular dynamics simulation of AR-perfluorooctanoic acid, ER-α-diethylstilbestrol, and ER-β-diethylstilbestrol were evaluated on LARMD server set to run at 1 ns in an explicit water model [24]. The root mean square deviation (RMSD), the radius of gyration (Rg), fraction of native contacts (Q), root mean square fluctuation (RMSF) and principal component analysis (PCA) of the ligand-protein complexes along the simulation trajectory were calculated and analysed. The binding free energy of these ligand-protein complexes was calculated using molecular mechanics Poisson–Boltzmann or Generalized Born surface area (MM-PB/GBSA) method as shown in Equation (4) [24].
(4)$$\Delta G_{bind}=\Delta E_{bind}-T\Delta S_{sol}-T\Delta S_{conf}$$

Δ*G_bind_*: binding free energy, Δ*E_bind_:* binding energy, *T*Δ*S_sol_*: solvation entropy, and *T*Δ*S_conf_*: conformational entropy.

### 2.4. Target Gene Expression and Phylogenetic Analysis

The upstream regulatory networks from signatures of differentially expressed genes obtained from toxicants’ target prediction were determined by transcription factor enrichment analysis, protein–protein interaction network expansion, and kinase enrichment analysis, using the predicted target genes on eXpression2Kinases (X2K) [10].

## 3. Result and Discussion

### 3.1. Toxicokinetic Prediction of Toxicants

The toxicants’ pharmacokinetic properties are shown in Figure 1, with perfluorooctanoic acid reported as an outlier due to its WLog P value of 10.75. Dimethyl phthalate, 4-bromodiphenylether, 4-nonylphenol, 4-octylphenol, 4-tert-butylphenol, bisphenol A, diethylstilbestrol, methoxychlor, tetrachloroethylene, and O-phenylphenol were predicted to have a high probability of permeating through the blood-brain barrier (BBB) and passively be absorbed by the gastrointestinal tract (GIT), while diethylhexyl phthalate and permethrin were both predicted to have high GIT but low BBB permeability. All other toxicants were not predicted to be either GIT or BBB permeants. All toxicants were predicted to be nonsubstrates of P-glycoprotein (Pgp) except diethylhexyl phthalate and tetrachlorodibenzodioxin.

In Table 3, all the toxicants inhibited one or more isoforms of cytochrome P_450_ (CYP_450_). Some of these toxicants’ solubility was previously reported, corroborating this finding [25]. This could further suggest these toxicants can easily be bioaccumulated in tissues where they can induce oxidative stress, insulin resistance, and elicit neurotoxicity [11]. The inhibition of CYP_450_ isoforms by these toxicants will lead to reduced metabolism of drugs, resulting in drug metabolism malfunctioning, drug resistance phenomenon, and elevated toxicity [20]. The synthetic accessibility score of these toxicants suggests that they are easily synthesised, available, and incorporated into different products. This accounts for the increased incorporation of these compounds in various products used daily by people [4].

### 3.2. Sex Hormone Receptor Activity of Toxicants

In SwissTargetPrediction, the similarity principle via reverse screening is the basis through which the probable protein targets of small molecules are predicted [14]. Dichlorodiphenyldichloroethylene, dichlorodiphenyldichloroethane, tetrachloroethylene, dimethyl phthalate, 4-bromodiphenylether, tetrachlorodibenzodioxin, and triphenyltin were not predicted to target any of the sex receptors. Estrogen receptor-α (ER-α) was predicted to be a highly probable target of 4-nonylphenol, 4-octylphenol, and diethylstilbestrol, while estrogen receptor-β (ER-β) was predicted to be a highly probable target of diethylstilbestrol. Diethylstilbestrol showed a high probability of targeting ER-α and -β, while bisphenol A showed a high probability of targeting all three receptors. All other toxicants had a weak probability of targeting one or more of the sex receptors (as illustrated in Table 4). ER-α, ER-β, and AR are important receptors involved in peripheral tissue glucose metabolism. They promote glucose and energy homeostasis upon androgen/estrogen binding in these tissues by reducing lipogenesis and increasing insulin secretion and sensitivity [7]. Since these receptors are targets of these toxicants, they are likely to alter glucose homeostasis and insulin release mechanism [4]. These toxicants were reported to be highly lipophilic; thus, they can permeate the membrane and bind to these receptors in the cytoplasm, nuclear membrane, or cell membrane [26].

### 3.3. Molecular Docking of Toxicants

Molecular docking is a conventional structure-based computational method used for analyzing binding interactions between protein–ligand complexes. In toxicity, this method is relevant in predicting the binding poses of toxicants in their various molecular targets’ binding pockets. The predicted activity between sex hormone receptors and environmental toxicants were previously reported in Table 4. Hence, Table 5 shows the docking scores of environmental toxicants in binding pockets of their predicted receptors of activity. From the docking scores shown in Table 5, the sex hormones (α-estradiol, β-estradiol, and dihydrotestosterone (DHT)) had high binding scores in the pocket of their respective receptors (ER-α, ER-β, and AR respectively) with bisphenol A and diethylstilbestrol exhibiting comparable affinity in the binding pocket of all three receptors. However, in AR’s binding pocket, high docking scores were recorded for dichlorodiphenyltrichloroethane, O-phenylphenol, and perfluorooctanoic acid, while in ER-β, a similar observation was made for permethrin (as illustrated in Table 5).

These high-binding affinities could suggest possible antagonism of these receptors either by competitive or mixed mechanism due to the benzene rings present in these toxicants’ structure [27]. The binding energy of the endogenous ligands in the receptors was lower than the toxicants as expected due to their role in the hormone system. This finding was similar to that reported by Jeong et al. [28], where the endogenous ligands had better docking scores than over 95% of the evaluated toxicants. The toxicants that their binding affinities were not determined would probably utilise different mechanisms other than binding to sex hormone receptors to elicit their toxic effect. Triphenyltin and tetrachlorodibenzodioxin were reported to bind to retinoid X receptor, peroxisome proliferator-activated receptor gamma (PPARγ), and aryl hydrocarbon receptor [29,30], while others such as tetrachloroethylene, dichlorodiphenyldichloroethane, and dichlorodiphenyldichloroethylene have neuronal nicotinic acetylcholine receptors [31] PPARγ, PPARα, and ryanodine receptors (type 1 and 2) [32,33,34] as their targets. The LE and LipE of the top-ranking toxicants were ≥0.3 and 3, respectively. However, the LipE of dichlorodiphenyltrichloroethane and perfluorooctanoic acid was less than 3 (as illustrated in Table 6). These values (0.3 and 3) were identified as the respective ideal threshold cutoff of LE and LipE. These values further buttress these toxicants’ effectiveness to bind to these receptors and elicit their toxic tendencies such as T2DM [25]. These toxicants were experimentally validated to exhibit estrogenic and androgenic properties in different model systems and cell lines [35,36,37]. In utero [38], perinatal [39] and late-life exposure [40,41] of rodents to BPA and DDT were reported to result in reduced insulin secretion and impaired glucose tolerance. Studies carried out by Lee et al. [42] and Wolf et al. [43] revealed that exposure to these environmental toxicants is associated with increased T2DM risk via endocrine disruption in humans. Some other reports presented a similar association in the elderly [44,45] and middle-aged women [46].

Similar to DHT interaction in AR binding pocket, bisphenol A and O-phenylphenol were stabilised by one hydrogen bond interaction with GLN711, while diethylstilbestrol hydrogen bond interaction was with MET745. On the other hand, dichlorodiphenyltrichloroethane and perfluorooctanoic acid exhibited 0 and 2 (ARG752 and THR877) hydrogen bond interactions, respectively, with AR (as illustrated in Figure 2).

Also, unlike other toxicants, perfluorooctanoic acid was stabilised by halogen bond interaction with 5 amino acid residues (LEU704, ASN705, GLN711, MET745, and PHE764) in AR binding pocket (as illustrated in Figure 2E). Halogen bonds are usually short and numerous, forming strong noncovalent interactions [47]; hence the multiple halogen interaction observed in the perfluorooctanoic acid–AR complex could be attributed to the low-binding energy observed. Halogen interactions were also identified to play a role in substrate specificity [48]. This could be a reason perfluorooctanoic acid was predicted to target only AR. Some of the amino acid residues such as LEU704, LEU707, MET742, and PHE764 were common π-π alkyl interactions stabilising DHT and the top-ranking toxicants, while ASN705 and GLY708 were the most common observed Van der Waals interactions with AR (as illustrated in Figure 2). Contrary to other toxicants, the amino acid residues involved in π-π alkyl and Van der Waals interactions between O-phenylphenol and AR were different (as illustrated in Figure 2D). This study further substantiates previous reports where LEU701, LEU704, ASN705, MET742, and PHE764 were amino acid residues in AR-binding pocket that interacted with toxicants [49,50].

The interaction of the top-ranking toxicants in Figure 3 depicted bisphenol A, diethylstilbestrol, and α-estradiol all interacting with GLU353 in ER-α binding pocket via hydrogen bond interaction, while ARG394 and MET421 were other hydrogen interactions with bisphenol A and α-estradiol, respectively. Common π-π alkyl (ALA350, LEU384, LEU384, and LEU391) and Van der Waals interactions (PHE425 and LEU428) were observed stabilising α-estradiol and the toxicants. It was also observed that diethylstilbestrol interacted with similar amino acid residues (MET343, LEU349, ARG394, GLY521, and LEU525) as α-estradiol. These interactions were not observed for bisphenol A, which accounted for the low binding energy recorded for diethylstilbestrol (as illustrated in Figure 3). In the binding pocket of ER-β, ARG346 was depicted as the common hydrogen bond interaction between the receptor, β-estradiol, and two toxicants (diethylstilbestrol and permethrin). However, other hydrogen interactions were observed for bisphenol A (GLU305 and HIS475) and diethylstilbestrol (MET295, LEU339, and HIS475).

Despite the difference in the amino acid residues involved in other binding interactions between β-estradiol and the toxicants, similar π-π alkyl (ALA302 and LEU343) and Van der Waals interactions (LEU298, GLU305, ILE373, PHE377 and GLY472) were identified in the binding pose of β-estradiol and diethylstilbestrol in ER-β (as illustrated in Figure 4). These common similarities and the increased number of hydrogen bond interaction with ER-β binding pocket amino acid residues could be attributed to the slightly lower binding energy observed for diethylstilbestrol. Comparing the binding pose of the toxicants and β-estradiol, permethrin had the least similarity. This observation might be ascribed to its structure as it can be seen overlapping out of ER-β binding pocket.

Different studies reported similar amino acid residues such as GLU353, ARG394, PHE404, and PHE425 that stabilise the binding of toxicants and EDCs in the active site of ER, which corroborate this study [25,51,52]. The side chains of GLU and ARG ensures upon orientation that these amino acid residues form hydrogen bonds with the ligands, while that of PHE is nonpolar, which makes it suitable for nonpolar interactions. Figure 5 shows the redocked ligands in comparison with that of the experimental pose after superimposition. The RMSD of the experimental and docked ligands in AR, ER-α, and ER-β pocket was 0.77, 0.85, and 0.61 Å, respectively. This low deviation value (<2.0 Å) credits the docking parameters rationality and docking method reliability used in this study. It further infers the toxicants’ strong affinity in the receptors’ pockets [53].

### 3.4. Molecular Dynamics of Toxicants

Protein dynamics is an important technique that gives greater insight into the protein’s dynamics at the atomic level and its relationship with the structure. In ligand-driven protein dynamics, the effect of the protein’s internal dynamics variation on ligand binding is the outcome [54]. The interactions between the top-ranking toxicants and their various docked complexes (AR-perfluorooctanoic acid, ER-α-diethylstilbestrol, and ER-β-diethylstilbestrol) were stable with <1.5 Å RMSD, while their radius of gyration (Rg) was within ±0.2 Å from the start point over the simulation timestep (as illustrated in Figure 6i,ii).

The obtained RMSD and Rg values would suggest the complexes formed were equilibrated and had low oscillation with a folded polypeptide structure that was relatively stable. It can also be hypothesised that there is no shift in ligand position as contrarily reported in a study by Ali [55], suggesting protein–ligand complex compactness. A steady decline in the fraction of native contact was observed from 1.0 to 0.97 (AR-Perfluorooctanoic acid and ER-β-Diethylstilbestrol) and 0.94 Q[X] (ER-α-Diethylstilbestrol) with RMSF fluctuations being dominant majorly at the loop region (as illustrated in Figure 6iii,iv). The larger fluctuations observed in the loops are normal as they are the most flexible protein structure regions. From the docking analysis, they are not involved in binding the toxicants; hence, they are surface loops, energetically favouring the decline in the fraction of native contact. These RMSF values of the helical structures suggest a little variation of the ligands from the initial binding pose after the MD simulation [56].

In AR-perfluorooctanoic acid PCA scree plot, PC1, PC2, and PC3 accounted for 19.2, 9.1, and 6.2% of the variance, respectively; subsequently, the individual PC’s contribution dropped below 4.3%. From the PCA scree plot for ER-α-diethylstilbestrol, PC1 accounts for more than one-quarter of the overall variance, strongly dominating the overall variance. When computed with the next 2 PC’s, they made up 41.9% of the variance. After the 4th PC, individual component contributions were below 3.5%. PC1 accounted for more than one-fifth of the overall variance, with PC2 and PC3 contributing a little over 10%, strongly dominating the overall variance. A drop below 5% was subsequently observed for other individual component contributions as depicted in ER-β-diethylstilbestrol PCA scree plot (as illustrated in Figure 6v). PC1, PC2, and PC3 retained most of the variance while providing a useful description of the original distribution’s atomic fluctuations. The large initial intensive motions rapidly declined to more localised fluctuations. The first 3 PC’s were used from the cluster analysis to analyse conformational transitions of the simulated systems by projecting their trajectories on a 2D subspace. The continuous colour scale, which changed from blue to white and then red, indicates that the conformational behavior of the three simulated systems exhibited no observable difference between the blue and red conformations as projected along the direction of PC2–PC3, while a difference along the direction of PC1–PC2 and PC1–PC3 between the 2 colour conformations were observed (as illustrated in Figure 6v). The continuous colour scale changed from blue to white and then red, indicating intermittent transitions between these conformations [57]. Electrostatic interactions contributed little, while Van der Waals interactions were the most contributors to the total free energy and binding free energy in all 3 complexes. The total free energy was counteracted by the entropy generated (as illustrated in Table 7). The difference in the proportion of variance observed in the principal component analysis (PCA) was evident in the predicted binding energy by MM-PB/GBSA (as illustrated in Table 7). Studies reported a positive correlation between MM-PB/GBSA predicted binding energy and experimental results [58,59]. The calculated MM-PB/GBSA binding energy corroborates the docking analysis that was earlier reported, suggesting these toxicants can interact with these receptors and elicit their toxic effects. The increased free binding energy observed in ER-β-diethylstilbestrol may be due to lower Van der Waals energy contribution [60].

### 3.5. Gene Expression Network Modified by Toxicants

The expression of genes, proteins, and transcription factors upregulated by toxicants upon sex hormone receptor binding from the gene network analysis is depicted in Figure 7a.

EGR1 (early growth response protein 1), ESR1 (estrogen receptor 1), VDR (vitamin D receptor), SUZ12 (polycomb protein), PPARD (peroxisome proliferator-activated receptors delta), NFE2L2 (nuclear factor erythroid 2-related factor 2), ZEB1 (zinc finger e-box binding homeobox 1), NFYB (nuclear transcription factor y subunit beta), and TP63 (tumour protein P63) were the transcription factors predicted to be induced by the toxicants upon protein–protein interaction. They interacted with mitogen-activated protein kinase (MAPK), casein kinase (CSNK), cyclin-dependent kinase (CDK), extracellular receptor kinase (ERK), homeodomain-interacting protein kinase (HIPK), tyrosine-protein kinase (ABL), and some intermediate proteins, namely PTEN, JUN, AKT1, NfKB1, MYC, RB1, SP1, SMAD3 and others as depicted in the interaction nodes. Some of these transcription factors, kinases, and intermediate proteins were previously identified as key genes and pathways implicated in T2DM prognosis and pathogenesis and potential pharmacological targets substantiating the result [61,62]. EGR1 was identified upon analysis as the most enriched transcription factor expressed by the toxicants followed by ESR1 and TP63, while MAPK1 was the most enriched kinase expressed by the toxicants followed by casein kinase 2 alpha 1 (CSNK2A1) and MAPK3 based on the hypergeometric *p*-value.

Upregulation of EGR1 in diabetes-related diseases was noted in numerous studies. It enhances extracellular matrix production and proliferation of mesangial cells by interacting with TGF-β1 and upregulation of downstream genes [63,64]. Glucose transporter type 1 (GLUT1) is a transcriptional target of TP63. Upon activation, there is an elevated expression of GLUT1, which leads to GLUT1-mediated glucose influx and generation of nicotinamide adenine dinucleotide phosphate (NADPH) from the pentose phosphate pathway (PPP) [65]. Upregulation of CSNK2A1 gene expression and protein concentration is associated with T2DM and its associated pathologies [66]. Upregulation of important intermediary proteins such as PTEN, AKT1, NfKB1, SP1, and SMAD3 was identified as a common occurrence in T2DM and its vascular complications [67,68,69]. Various experimental studies showed that binding of environmental toxicants to sex hormone receptors led to increased AKT and CREB phosphorylation and increased PI3K and NfKβ activity and MAPK and ERK activation while inhibiting PPAR, corroborating the results [70,71,72,73,74,75,76,77]. These molecular activities lead to insulin action impairment and intermediary metabolism alteration [70,78,79].

## 4. Conclusions

This study identified some daily used environmental toxicants and plasticizers (such as bisphenol A, diethylstilbestrol, and others) that utilize sex hormone receptors as their metabolic target. These toxicants bind tightly to these sex receptors with binding scores comparable to that of the endogenous substrates of these receptors while exhibiting similar interactions. Dynamics of the protein-ligand structures further depicted stable binding and equilibrated complexes between these toxicants and the sex receptors. The binding may alter the downstream metabolic function of the receptors by upregulating certain gene signatures such as MAPK, TP63, ERK, PTEN, AKT1, and NfKB1, which are key transcription factors and kinases implicated in T2DM prognosis and pathogenesis. They were also identified as playing a role in energy and intermediary metabolism. Despite these findings, molecular docking using different docking algorithms and running multiple replicas of the protein-ligand complex dynamics could be done to avoid making false-positive conclusions. Regardless of these limitations and future directions, plasticizers and pharmaceuticals are the major products with these toxicants. Hence, monitoring how these toxicants are manufactured, used, and disposed of should be enforced. It is necessary to ensure reduced interaction between various communities and these toxic chemicals, thereby reducing the incidence and prevalence of type 2 diabetes mellitus.

## Figures and Tables

**Figure 1 toxics-09-00255-f001:**
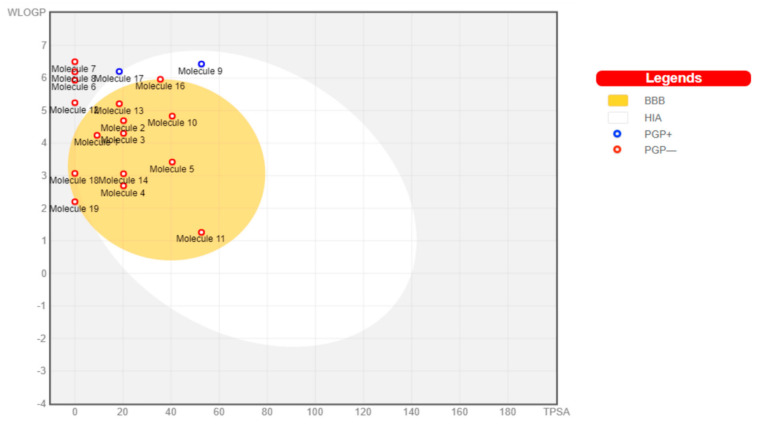
Boiled egg graph of toxicants showing predicted pharmacokinetic properties. HIA: human intestinal absorption, BBB: blood-brain barrier, PGP: p-glycoprotein, +: substrate, −: nonsubstrate.

**Figure 2 toxics-09-00255-f002:**
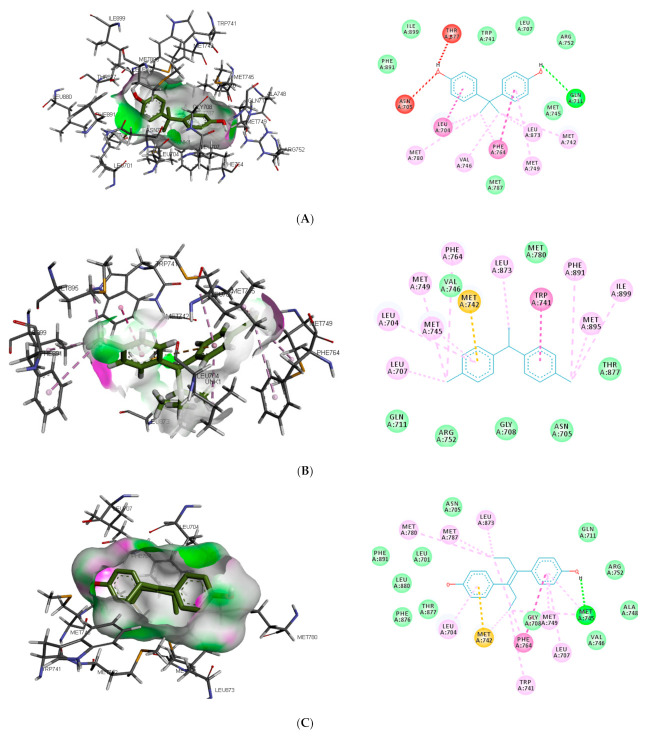
3D and 2D binding mode of (**A**) bisphenol A, (**B**) dichlorodiphenyltrichloroethane, (**C**) diethylstilbestrol, (**D**) O-phenylphenol, (**E**) perfluorooctanoic acid, and (**F**) dihydrotestosterone in affinity with androgen receptor. Hydrogen, halogen, unfavourable, and π bonds are depicted as green, luminous blue, red, and any other coloured (purple, magenta, orange, and pink) lines, respectively, while Van der Waals interactions appear as light green circles.

**Figure 3 toxics-09-00255-f003:**
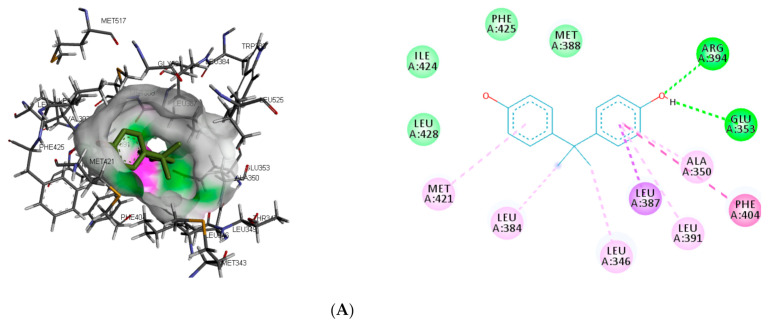
3D and 2D binding mode of (**A**) bisphenol A, (**B**) diethylstilbestrol and (**C**) α-estradiol in affinity with estrogen receptor-α. Hydrogen and π bonds are depicted as green and any other coloured (purple, magenta, and pink) lines, respectively, while Van der Waals interactions appear as light green circles.

**Figure 4 toxics-09-00255-f004:**
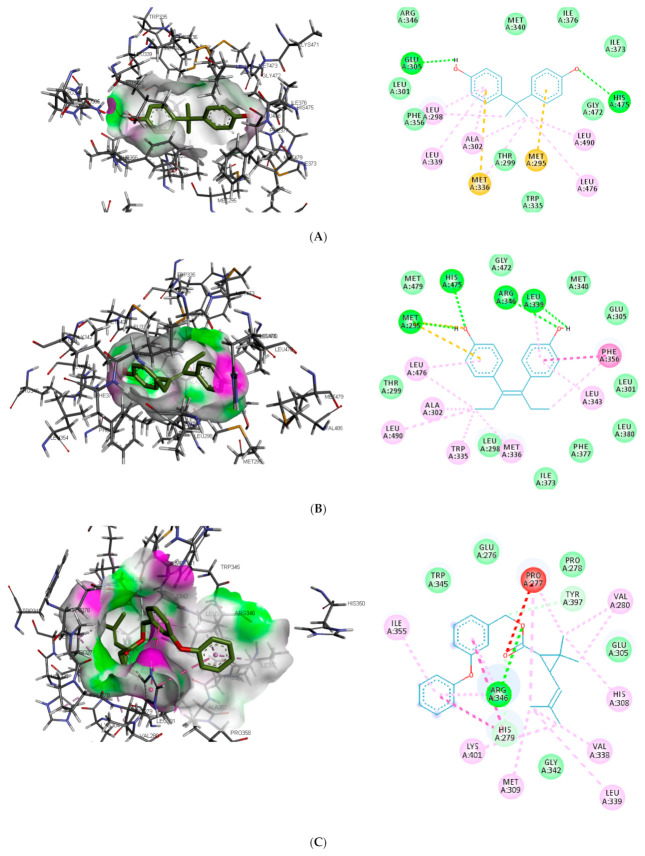
3D and 2D binding mode of (**A**) bisphenol A, (**B**) diethylstilbestrol, (**C**) permethrin and (**D**) β-estradiol in affinity with estrogen receptor-β. Hydrogen, carbon-hydrogen, unfavourable, and π bonds are depicted as green, pale blue, red, and any other coloured (magenta, orange, and pink) lines, respectively, while Van der Waals interactions appear as light green circles.

**Figure 5 toxics-09-00255-f005:**
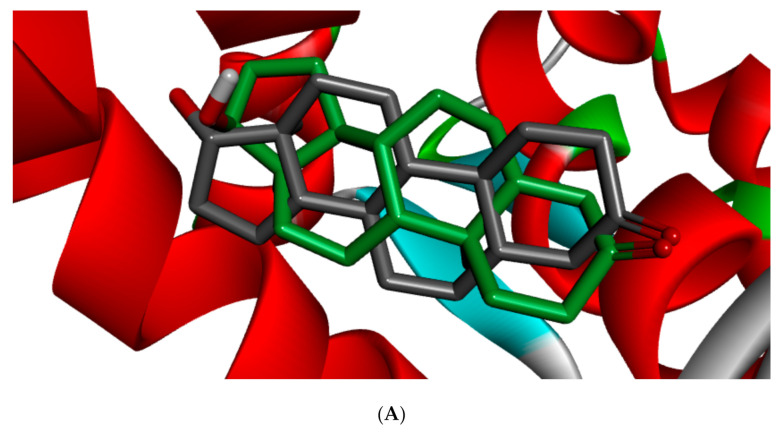
3D binding mode and superimposition of experimental (grey ligand) and docking (green and brown ligand) pose in affinity with (**A**) 1T5Z, (**B**) 5KRJ, and (**C**) 5TOA.

**Figure 6 toxics-09-00255-f006:**
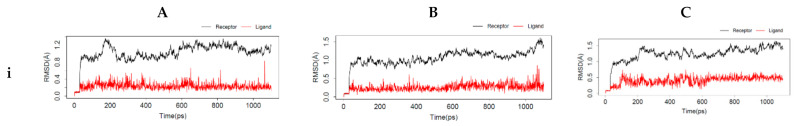
Molecular dynamics simulation of (**A**) androgen receptor-perfluorooctanoic acid, (**B**) estrogen receptor-α-diethylstilbestrol, and (**C**) estrogen receptor-β-diethylstilbestrol showing (**i**) root mean square deviation, (**ii**) radius of gyration, (**iii**) fraction of native contacts, (**iv**) root mean square fluctuation, and (**v**) principal component analysis.

**Figure 7 toxics-09-00255-f007:**
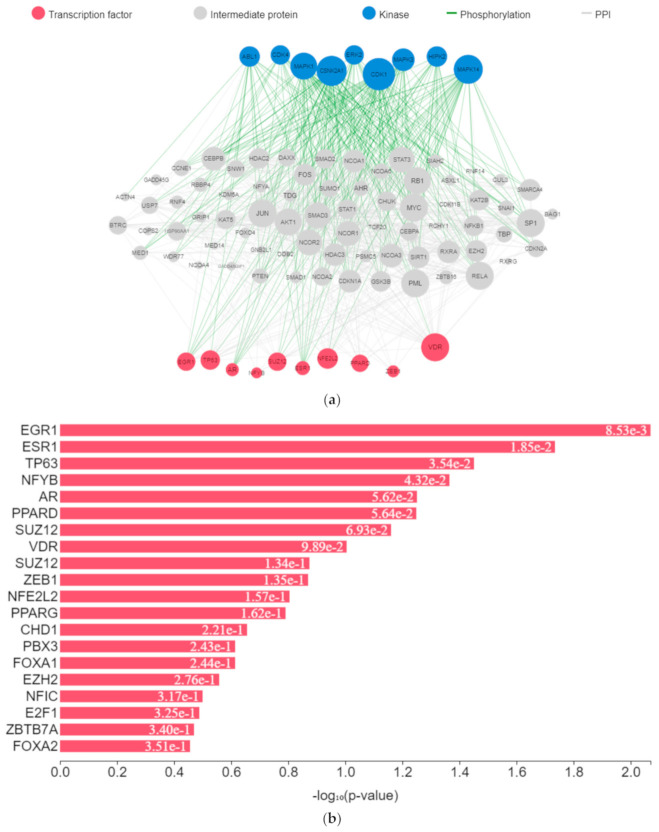
(**a**) Expression network of target transcription factors and kinases genes upregulated by toxicants. (**b**) Transcription factor enrichment analysis for toxicants target genes. (**c**) Kinase enrichment analysis for toxicants’ target genes.

**Table 1 toxics-09-00255-t001:** List of selected environmental toxicants associated with sex hormone receptors implicated in type 2 diabetes mellitus (T2DM).

Mol No.	Environmental Toxicants	SMILES
1	4-bromodiphenylether	C1=CC=C(C=C1)OC2=CC=C(C=C2)Br
2	4-nonylphenol	CCCCCCCCCC1=CC=C(C=C1)O
3	4-octylphenol	CCCCCCCCC1=CC=C(C=C1)O
4	4-tert-butylphenol	CC(C)(C)C1=CC=C(C=C1)O
5	Bisphenol A	CC(C)(C1=CC=C(C=C1)O)C2=CC=C(C=C2)O
6	Dichlorodiphenyldichloroethane	C1=CC(=CC=C1C(C2=CC=C(C=C2)Cl)C(Cl)Cl)Cl
7	Dichlorodiphenyltrichloroethane	C1=CC(=CC=C1C(C2=CC=C(C=C2)Cl)C(Cl)(Cl)Cl)Cl
8	Dichlorodiphenyldichloroethylene	C1=CC(=CC=C1C(=C(Cl)Cl)C2=CC=C(C=C2)Cl)Cl
9	Diethylhexyl phthalate	CCCCC(CC)COC(=O)C1=CC=CC=C1C(=O)OCC(CC)CCCC
10	Diethylstilbestrol	CCC(=C(CC)C1=CC=C(C=C1)O)C2=CC=C(C=C2)O
11	Dimethyl phthalate	COC(=O)C1=CC=CC=C1C(=O)OC
12	Heptachlor	C1=CC(C2C1C3(C(=C(C2(C3(Cl)Cl)Cl)Cl)Cl)Cl)Cl
13	Methoxychlor	COC1=CC=C(C=C1)C(C2=CC=C(C=C2)OC)C(Cl)(Cl)Cl
14	O-phenylphenol	C1=CC=C(C=C1)C2=CC=CC=C2O
15	Perfluorooctanoic acid	C(=O)(C(C(C(C(C(C(C(F)(F)F)(F)F)(F)F)(F)F)(F)F)(F)F)(F)F)O
16	Permethrin	CC1(C(C1C(=O)OCC2=CC(=CC=C2)OC3=CC=CC=C3)C=C(Cl)Cl)C
17	Tetrachlorodibenzodioxin	C1=C2C(=CC(=C1Cl)Cl)OC3=CC(=C(C=C3O2)Cl)Cl
18	Tetrachloroethylene	C(=C(Cl)Cl)(Cl)Cl
19	Triphenyltin	C1=CC=C(C=C1)[Sn](C2=CC=CC=C2)C3=CC=CC=C3

Mol No.: Molecule Number.

**Table 2 toxics-09-00255-t002:** Grid box coordinates and size parameters used for molecular docking.

Dimension	Dimension (Å)			Centre (Å)		
	AR	ER-α	ER-β	AR	ER-α	ER-β
x	22	20	26	4	13	20
y	26	20	24	60	0	40
z	24	24	26	5	4	13

AR: Androgen receptor; ER: Estrogen receptor.

**Table 3 toxics-09-00255-t003:** Predicted metabolic profile of toxicants.

Environmental Toxicants	Cytochrome P_450_ Isoform Inhibition					SP (cm.s^−1^)	BA	SA
	1A2	2C19	2C9	2D6	3A4			
4-bromodiphenylether	Yes	Yes	Yes	No	No	−4.77	0.55	1.81
4-nonylphenol	Yes	Yes	No	Yes	No	−3.55	0.55	1.63
4-octylphenol	Yes	No	No	Yes	No	−3.85	0.55	1.53
4-tert-butylphenol	Yes	No	No	No	No	−4.87	0.55	1
Bisphenol A	Yes	No	No	Yes	No	−5.34	0.55	1.43
Dichlorodiphenyldichloroethane	Yes	Yes	Yes	Yes	No	−3.98	0.55	2.37
Dichlorodiphenyldichloroethylene	No	Yes	Yes	No	No	−3.62	0.55	2.08
Dichlorodiphenyltrichloroethane	Yes	Yes	Yes	No	No	−3.56	0.55	2.37
Diethylhexyl phthalate	No	No	Yes	No	Yes	−3.39	0.55	4.12
Diethylstilbestrol	Yes	Yes	Yes	Yes	Yes	−4.34	0.55	2.35
Dimethyl phthalate	Yes	No	No	No	No	−6.35	0.55	1.55
Heptachlor	Yes	Yes	Yes	No	No	−5.49	0.55	4.75
Methoxychlor	Yes	Yes	Yes	Yes	No	−4.8	0.55	2.43
O-phenylphenol	Yes	Yes	Yes	No	No	−5.14	0.55	1.27
Perfluorooctanoic acid	No	Yes	Yes	No	No	−5.33	0.85	2.47
Permethrin	Yes	Yes	Yes	Yes	Yes	−4.07	0.55	3.73
Tetrachlorodibenzodioxin	No	No	Yes	No	No	−3.44	0.55	2.71
Tetrachloroethylene	No	No	No	No	No	−4.9	0.55	2.59
Triphenyltin	Yes	No	No	Yes	No	−4.66	0.55	1.81

SP, BA, and SA represent skin permeation, bioavailability, and synthetic accessibility, respectively.

**Table 4 toxics-09-00255-t004:** Predicted activity between sex hormone receptors and environmental toxicants.

Environmental Toxicants	Androgen Receptor	Estrogen Receptor-α	Estrogen Receptor-β
4-bromodiphenylether	-	-	-
4-nonylphenol	*	*****	*
4-octylphenol	*	*****	*
4-tert-butylphenol	*	*	*
Bisphenol A	*****	*****	*****
Dichlorodiphenyldichloroethane	-	-	-
Dichlorodiphenyldichloroethylene	-	-	-
Dichlorodiphenyltrichloroethane	*	-	-
Diethylhexyl phthalate	*	-	-
Diethylstilbestrol	*	*****	*****
Dimethyl phthalate	-	-	-
Heptachlor	*	-	-
Methoxychlor	*	*	*
O-phenylphenol	*	*	*
Perfluorooctanoic acid	*	-	-
Permethrin	-	-	*
Tetrachlorodibenzodioxin	-	-	-
Tetrachloroethylene	-	-	-
Triphenyltin	-	-	-

- (0%), * (1–20%), ** (21–40%), *** (41–60%), **** (61–80%), ***** (81–100%) probability of receptor binding.

**Table 5 toxics-09-00255-t005:** Docking scores (kcal/mol) of environmental toxicants in binding pockets of receptors.

Environmental Toxicants	Androgen Receptor	Estrogen Receptor-α	Estrogen Receptor-β
4-nonylphenol	−7.1	−5.8	−6.9
4-octylphenol	−6.6	−5.9	−7.0
4-tert-butylphenol	−6.3	−5.9	−6.3
Bisphenol A	**−8.3**	**−7.7**	**−7.9**
Dichlorodiphenyltrichloroethane	**−7.9**	ND	ND
Diethylhexyl phthalate	−6.4	ND	ND
Diethylstilbestrol	**−8.6**	**−8.5**	**−8.6**
Heptachlor	−5.9	ND	ND
Methoxychlor	−6.0	−7.0	−5.4
O-phenylphenol	**−7.8**	−7.0	−7.4
Perfluorooctanoic acid	**−8.9**	ND	ND
Permethrin	ND	ND	**−7.9**
α-estradiol	ND	**−9.1**	ND
β-estradiol	ND	ND	**−8.5**
Dihydrotestosterone	**−9.2**	ND	ND

ND: Not determined. Binding energy in bold signifies compounds that are past the threshold of −7.5 kcal/mol.

**Table 6 toxics-09-00255-t006:** Binding effectiveness of top-ranking environmental toxicants in binding pockets of receptors.

Environmental Toxicants	Androgen Receptor			Estrogen Receptor-α			Estrogen Receptor-β		
	LE ^a^	BEI	LipE	LE ^a^	BEI	LipE	LE ^a^	BEI	LipE
Bisphenol A	0.49	40	6	0.45	38	5	0.46	39	5
Dichlorodiphenyltrichloroethane	0.42	25	2	-	-	-	-	-	-
Diethylstilbestrol	0.43	35	4	0.43	34	4	0.43	35	4
O-phenylphenol	0.60	51	6	-	-	-	-	-	-
Perfluorooctanoic acid	0.36	23	1	-	-	-	-	-	-
Permethrin	-	-	-	-	-	-	0.30	23	3

LE: ligand efficiency, BEI: binding efficiency index, LipE: lipophilic efficiency, ^a^: kcal/mol.

**Table 7 toxics-09-00255-t007:** Predicted molecular mechanics Poisson–Boltzmann (Generalised Born) surface area (MM-PB/GBSA) binding energy of (**A**) androgen receptor-perfluorooctanoic acid, (**B**) estrogen receptor-β-diethylstilbestrol, and (**C**) estrogen receptor-β-diethylstilbestrol.

	kcal/mol									
	EE	VdW	G	PB_SOL_	PB_TOT_	GB_SOL_	GB_TOT_	TS	ΔPB	ΔGB
A	−1.60	−36.96	−38.56	8.99	−29.57	1.76	−36.80	16.72	−12.85	−20.08
B	−2.68	−41.75	−44.43	12.23	−32.20	4.44	−39.99	19.67	−12.53	−20.32
C	−1.73	−51.66	−53.39	18.42	−34.97	7.23	−46.16	17.45	−17.52	−28.71

EE: Electrostatic energy, VDW: Van der Waals contribution, G: total gas phase energy, PB_SOL_: nonpolar solvation contribution, GB_SOL_: polar solvation contributions, TS: entropy, PB_TOT_: Poisson–Boltzmann total energy, GB_TOT_: Generalised born total energy, ΔPB/ΔGB: Final estimated binding free energy.

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
