# Peer review of "Type 2 Diabetes Mellitus Mediation by the Disruptive Activity of Environmental Toxicants on Sex Hormone Receptors: In Silico Evaluation"

_toxics, 2021, doi:10.3390/toxics9100255_

Round 1

Reviewer 1 Report

The manuscript of Iheagwam and co-authors investigate the effect of environmental toxicants on androgen and alpha/beta estrogen receptors mediating type 2 diabetes mellitus.
The work as presented it is not mature for a publication, there are many lacking information in the methodological part and also the computational protocols do not fulfill the statistical robustness to support an entire in silico investigation (for example running different independent replicas or having longer MD simulations). Also, the description of the results is not exhaustive in all the findings and data presented with a lot of redundancy in the presentation of the images.

Author Response

Reviewer 1

The work as presented it is not mature for a publication, there are many lacking information in the methodological part and also the computational protocols do not fulfill the statistical robustness to support an entire in silico investigation (for example running different independent replicas or having longer MD simulations). Also, the description of the results is not exhaustive in all the findings and data presented with a lot of redundancy in the presentation of the images

The methodology has been recast to show more information. For the fulfilment of statistical robustness (eg. longer MD simulations), the webserver used to run this simulation has 1 ns as its default parameter and different publications have accepted this method (https://www.tandfonline.com/doi/full/10.1080/07391102.2020.1794971; https://www.tandfonline.com/doi/full/10.1080/07391102.2020.1813630).

The description of the results has been made exhaustive in all data and findings presented. The redundancy in the presentation of the images have been taken care of.

Reviewer 2 Report

It is not stated why those 19 compounds were selected for this study.

Molecule numbering in Table 1 should be corrected.

Line 82: grid spacing in Vina is 1 Angstrom by default, and can't be set.

Why in table 5 only a part of the docking results are presented? The "ND" entries should be discussed in the text.

Line 179: amino acid info missing "425"

Line 181:  what is LEU701/4?

Regarding the docking result:

- the cartoon representations in figure 2-5 should be replaced with a proper line/stick representation of the binding site, and the involved residues should be highlighted. If 2D interaction diagrams are kept, then do not use the same color for different interactions. Perhaps this images should go in the SPI.

- binding interactions between receptor residues and molecules should be discussed in detail in the text.

Each crystal structure contains a bound ligand, redocking should be performed in each case for validation.

The conclusions section must be improved.

I would suggest that the authors use the molecule number in the text, instead of the names, to be easier to follow.

The docking results should be correlated with some experimental results!

I would replace the word "buttresses"

Author Response

Reviewer 2

It is not stated why those 19 compounds were selected for this study.

These 19 compounds which were selected from literature as highlighted in the text were the most commonly used components of different products used worldwide. These toxicants also cut across all the classifications of toxicants from bisphenols, phthalates and their esters, diethylstilbestrol, polychlorinated biphenyls, dioxins and others.                

Molecule numbering in Table 1 should be corrected

Molecule numbering in Table 1 is correct

Line 82: grid spacing in Vina is 1 Angstrom by default, and can't be set.

Grid box was set in autodock 4.2 and the grid spacing is 0.375 by default.

Why in table 5 only a part of the docking results are presented? The "ND" entries should be discussed in the text.

The predicted activity between sex hormone receptors and environmental toxicants have previously been reported in table 4, hence, table 5 shows the docking scores of environmental toxicants in binding pockets of their predicted receptors of activity.

Line 179: amino acid info missing "425"

Amino acid info has been added

Line 181:  what is LEU701/4?

The correct presentation format has been inputed

The cartoon representations in figure 2-5 should be replaced with a proper line/stick representation of the binding site, and the involved residues should be highlighted. If 2D interaction diagrams are kept, then do not use the same color for different interactions. Perhaps this images should go in the SPI.

In the cartoon representation, ribbons were used and the involved residues were highlighted. Using line/stick representation of the binding site, and the involved residues is just a re-representation of the 2D interaction diagram. The authors will also like to note that different colours were used for different interactions in the 2D diagrams and each colour explicitly explained.

binding interactions between receptor residues and molecules should be discussed in detail in the text.

The binding interactions between receptor residues and molecules have been discussed in detail in the text

Each crystal structure contains a bound ligand, redocking should be performed in each case for validation.

Validation was carried out using 1t5z as stated in the methodology

I would suggest that the authors use the molecule number in the text, instead of the names, to be easier to follow.

The molecule number was just for easy identification of the boiled egg graph. The authors will prefer the names of the toxicants as it is easily relatable to readers and researchers alike.

The conclusions section must be improved.

The conclusion has been improved

The docking results should be correlated with some experimental results!

The docking results has been correlated with some experimental results

I would replace the word "buttresses"

It has been replaced with substantiates

Reviewer 3 Report

The authors present an interesting integrated computational approach to study the effect of environmental toxicant on type 2 diabetes mellitus mediated by the interaction with hormone receptors.

Although different analyses with different methodologies are reported, how the various pieces of evidence integrate with each other and ultimately contribute to prove the final thesis must be better explained, both in the description of the results of the different analyses carried out and in the conclusions section.

Please, consider the following:

Line 15: In the abstract chemicals “permithrin, perfluorooctanoic acid, bisphenol A and diethylstilbestrol” are cited as active toxic compounds, whereas in the text more toxicants are considered.

Line 15: please correct “permithrin” with “permethrin”

Line 37: The sentence is not clear to me, please consider using a verb other than "contraindicated"

Line 66: Table 1, please correct the molecule numbers (that should be consistent with the one reported in boiled egg graph)

Line 75: In my opinion this section is a bit too concise. Because there are many different methodologies, I think a few more details could be useful.

Line 77: PDB structure 5KRJ, seem to contain a different ligand (OBHS derivative). The following analyses with alpha estradiol ligand, how were they carried out?

Line 88: Was the "re-docking" of the original ligand done only for AR? In Table 5 docking scores are reported also for alpha and beta estradiol.

Line 88: which is the AR original ligand? A generic “androgen” is reported in the text. Perhaps it might be convenient to report the ligand of the 1T5Z structure, dihydrotestosterone (DHT)

Line 98: pIC50, Ki and Kd, please use the extended definitions the first time they occur.

Line 120: please correcr “illicit”

Line 128: Please provide definition for HIA acronym (human intestinal absorption), reported in the graph.

Line 131: How can the results of this section be interpreted in the light of the analyses that follow? Please, clarify.

Line 154: Please correct “affinity” with “affinities”

Line 155: The argument “due to the benzene rings present” does not seem to fit the case. On the one hand, not all "high bonding affinity" toxicants have a benzene ring (e.g. Perfluorooctanoic acid), on the other hand many of the low binding affinity chemicals have one.

Line 161-165: It is my opinion that the concept expressed in not fully clear.

Line 170 and following: I think that the description of the interactions in the binding pockets is a bit confusing. Which amino acid is reported for which receptor and toxicant or original ligand? Furthermore, they are not always identifiable in the figures. Comparison with “different studies” also seem not fully consistent.

Line 172: please specify, e.g., hydrogen bonding, π–π stacking and van der Waals interactions

Line 176: please correct “ight”

Line 177: please correct ”asribed”

Line 198: Legends to figures 2, 3, 4: color codes are not clearly reported. Please, consider revise.

Line 207: Molecular dynamics simulations were performed for which complexes? Please, clarify.

Line 221 and following: I think that PCA and predicted binding energy analysis require a more in-depth explanation. The results don't seem so easy to interpret. A brief description of the methodologies should also be included in the methods section.

Line 238: Gene Expression Network Modified by Toxicants, which toxicants have been considered? Please clarify.

Line 275: A summary of the evidence gathered in support of the thesis could be very helpful to report here in the conclusions section.

Author Response

Reviewer 3

Line 15: In the abstract chemicals “permithrin, perfluorooctanoic acid, bisphenol A and diethylstilbestrol” are cited as active toxic compounds, whereas in the text more toxicants are considered.

Dichlorodiphenyltrichloroethane and O-phenylphenol which were initially missing have been added

Line 15: please correct “permithrin” with “permethrin”

Done

Line 37: The sentence is not clear to me, please consider using a verb other than "contraindicated"                 

Contraindicated has been replaced with implicated

Line 66: Table 1, please correct the molecule numbers (that should be consistent with the one reported in boiled egg graph)

The molecule numbers are already consistent with the one reported in boiled egg graph

Line 75: In my opinion this section is a bit too concise. Because there are many different methodologies, I think a few more details could be useful.

The authors humbly disagree. In this section, only the molecular docking protocol was concise as there has been numerous details in various publications. All other methodologies such as protein abstraction, minimisation, molecular dynamics simulation, metrics of binding effectiveness and toxic bioactivity were well explained

Line 77: PDB structure 5KRJ, seem to contain a different ligand (OBHS derivative). The following analyses with alpha estradiol ligand, how were they carried out?

In the publication by Nwachukwu et al 2017 www.doi.org/10.1016/j.chembiol.2016.11.014, the effects of environmental estrogens and anti-hormone therapies on ERα cellular response was studied. OBHS derivative ligand in 5KRJ was one of the many studied ligands which was deposited with ERα. α-estradiol is ERα’s endogenous ligand and it was downloaded from pubchem and used for docking.

Line 88: Was the "re-docking" of the original ligand done only for AR? In Table 5 docking scores are reported also for alpha and beta estradiol.

Redocking was done for only AR as stated in the methodology

Line 88: which is the AR original ligand? A generic “androgen” is reported in the text. Perhaps it might be convenient to report the ligand of the 1T5Z structure, dihydrotestosterone (DHT).

Androgen has been replaced with Dihydrotestosterone

Line 98: pIC50, Ki and Kd, please use the extended definitions the first time they occur.

Done

Line 120: please correct “illicit”

Done

Line 128: Please provide definition for HIA acronym (human intestinal absorption), reported in the graph.

This has been done as well as other acronyms

Line 131: How can the results of this section be interpreted in the light of the analyses that follow? Please, clarify.

This results goes ahead to further show the sex hormone receptors that are likely targets of these environmental toxicant

Line 154: Please correct “affinity” with “affinities”

Done

Line 155: The argument “due to the benzene rings present” does not seem to fit the case. On the one hand, not all "high bonding affinity" toxicants have a benzene ring (e.g. Perfluorooctanoic acid), on the other hand many of the low binding affinity chemicals have one.

The emphasis is on the benzene rings which reduces the polarity of these compounds. Many of the low binding affinity chemicals have only one benzene ring while all the "high bonding affinity" toxicants bar Perfluorooctanoic acid have 2 benzene rings.

Line 161-165: It is my opinion that the concept expressed in not fully clear.

The concept of binding effectiveness has been recast for better clarity.

Line 170 and following: I think that the description of the interactions in the binding pockets is a bit confusing. Which amino acid is reported for which receptor and toxicant or original ligand? Furthermore, they are not always identifiable in the figures. Comparison with “different studies” also seem not fully consistent.

The binding interactions between receptor residues and molecules have been discussed in detail in the text

Line 172: please specify, e.g., hydrogen bonding, π–π stacking and van der Waals interactions

Binding interactions have been specified

Line 176: please correct “ight”

Line 177: please correct ”asribed”

Done

Line 198: Legends to figures 2, 3, 4: color codes are not clearly reported. Please, consider revise.

The authors humbly disagree with the reviewer as each colour coded line has been described for the interactions as well as the hydrophobic interactions.

Line 207: Molecular dynamics simulations were performed for which complexes? Please, clarify.

The complexes have been listed out for better clarification.

Line 221 and following: I think that PCA and predicted binding energy analysis require a more in-depth explanation. The results don't seem so easy to interpret. A brief description of the methodologies should also be included in the methods section.

In depth explanation has been given as regards PCA and predicted binding energy analysis. A brief description of the methodologies has also been included in the methods section.

Line 238: Gene Expression Network Modified by Toxicants, which toxicants have been considered? Please clarify.

The expression network is based on the inputed genes of the sex hormone receptors. Binding of the toxicants to these receptors will alter these genes as well as a host of kinases, proteins and transcription factors which are connected via intermediary metabolism altering normal metabolic processes such as glucose homeostasis and insulin response which are key to mediating diabetes mellitus.

A summary of the evidence gathered in support of the thesis could be very helpful to report here in the conclusions section.

The conclusion has been improved

Reviewer 4 Report

In general, the work was carried out at a good methodological level with the involvement of at least three methods of computer modeling. The research topic is relevant.
But I had a few comments regarding the design and presentation of the manuscript.
1. The authors presented 3 figures based on the results of molecular docking of biologically active substances in the active centers of sex hormones (Figures 2-4). But in my opinion, the presentation of the material in them is not entirely successful. In particular, it is not clear from these figures whether the studied structures are positioned into single clusters or are located in different areas of the active center. Alternatively, I propose to present each of Figures 2-4 as follows. Figure 2 can be prepared, in which all these five structures of biologically active substances are superimposed on each other in a single cluster (if so) or in different areas of space. To do this, you just need to superimpose the inhibitor structures on top of each other. And drawings that reflect the positioning of each individual structure in the active center of the androgen receptor can be placed in the Supplementary Materials. But I do not insist on my wish.
2. For Figures 3 and 4, the remarks are similar.
3. The axis labels and the names of the diagrams shown in Figure 6 are small and indistinct.
4. In conclusion, I would like to see the conclusions in a more detailed form.

Author Response

Reviewer 4

The authors presented 3 figures based on the results of molecular docking of biologically active substances in the active centers of sex hormones (Figures 2-4). But in my opinion, the presentation of the material in them is not entirely successful. In particular, it is not clear from these figures whether the studied structures are positioned into single clusters or are located in different areas of the active center. Alternatively, I propose to present each of Figures 2-4 as follows. Figure 2 can be prepared, in which all these five structures of biologically active substances are superimposed on each other in a single cluster (if so) or in different areas of space. To do this, you just need to superimpose the inhibitor structures on top of each other. And drawings that reflect the positioning of each individual structure in the active center of the androgen receptor can be placed in the Supplementary Materials. But I do not insist on my wish.

The authors appreciate this idea, however, it will be an extra re-representation of the results.

For Figures 3 and 4, the remarks are similar.

Response is similar as presented above

The axis labels and the names of the diagrams shown in Figure 6 are small and indistinct.

The axis labels are not small but rather reduced in size. When expanded upon publication, it will be visible as diagrams are >300dpi

In conclusion, I would like to see the conclusions in a more detailed form

The conclusion has been improved

Round 2

Reviewer 1 Report

Despite small ameliorations in the methodological description, the authors did not undergo to a complete revision of the manuscript, especially the points previously raised.

There are one major points about the statistical significance of the molecular dynamic trajectories. The author should consider longer trajectories or running multiple replicas.

Also other minor points:

  • HA (heavy atoms) instead of HE in equation 1 describing the Ligand Efficiency.
  • LE, LLE and BEI are not used as metric to assess toxic bioactivities (needs to be changed in the main text).
  • Figure 2, 3 and 4 has not been changed based on previous reviewer suggestions.

Author Response

SN

Reviewers Comment

Authors Response

Reviewer 1

1.       

There are one major points about the statistical significance of the molecular dynamic trajectories. The author should consider longer trajectories or running multiple replicas.

The authors used the default trajectory setting on the LARMD webserver to run MD. Running multiple replicas cannot be done on the LARMD server. Below are some publications that used similar protocol.

https://doi.org/10.2174/1570180817999200831094703

http://doi.org/10.5455/javar.2021.h481

https://doi.org/10.1155/2020/5324560

https://doi.org/10.1080/07391102.2020.1813630

https://doi.org/10.1080/07391102.2020.1794971

However, the authors have noted this valid point raised by the reviewer and have included them as limitations of the study.

Reviewer 2 Report

I will highlight some of my previous comments, which the authors failed to address:

1 - A paper that presents docking results, must include proper high quality images, that clearly shows the binding pose and the interactions with the corresponding residues. This is a must if someone wants to reproduce the results. The authors failed to deliver my request to improve figures 2-4.

Furthermore, I asked to use different colors in 2D interaction diagrams - green and light green are not that distinct!

2 - Line 77 and 78: AutoDock 4.2 is used for docking, not for preparing ligands/proteins (my guess is, the authors used ADTools).

"remove water and heteroatoms from protein structures ..." - this is nonsense

3 - the authors failed to discuss why many binding affinities were not determined in Table 5.

4 - the authors did not give any explanation why docking procedure was validated for only one crystal structure. Did the validation fail for the other cases?

5 - where are the correlations with the experimental results, which the authors claim to have been included in the text?

6 - the conclusion section is still very general, doesn't highlight the major findings.

7 - it should be "Van der Waals" not "waal"

Author Response

Reviewer 2

1.       

A paper that presents docking results, must include proper high quality images, that clearly shows the binding pose and the interactions with the corresponding residues. This is a must if someone wants to reproduce the results. The authors failed to deliver my request to improve figures 2-4.

Figure 2, 3 and 4 has been changed      

2.       

Furthermore, I asked to use different colors in 2D interaction diagrams - green and light green are not that distinct!

The authors disagree with the reviewer as green and light green are distinct!. Also, the light green circles do not have broken lines

3.       

Line 77 and 78: AutoDock 4.2 is used for docking, not for preparing ligands/proteins (my guess is, the authors used ADTools).

The omitted tools have been added to autodock

4.       

"remove water and heteroatoms from protein structures ..." - this is nonsense

The authors do not appreciate any part of this work being attributed as nonsense and hence will prefer the reviewer constructively criticize and make clear his/her point.

Removing water and heteroatoms from protein structures are processes that clean the protein into a nascent structure. Below are some few publications where this process was done.

https://doi.org/10.1371/journal.pone.0241773 https://doi.org/10.1016/j.colsurfb.2019.110640

https://doi.org/10.1016/j.molstruc.2019.05.104

https://doi.org/10.1016/j.poly.2021.115164

5.       

the authors failed to discuss why many binding affinities were not determined in Table 5

The reason why many binding affinities were not determined in Table 5 have been discussed.

To further buttress, they were not predicted to target the receptors where ND was allocated to them. Molecular docking was carried out only on toxicants and their predicted target receptors

6.       

the authors did not give any explanation why docking procedure was validated for only one crystal structure. Did the validation fail for the other cases?

All other docking parameters have been verified as shown in figure 5 b and c

7.       

where are the correlations with the experimental results, which the authors claim to have been included in the text?

It was situated just before table 5. However, other experimental results have been added

8.       

the conclusion section is still very general, doesn't highlight the major findings

The major findings the conclusion have been highlighted

9.       

it should be "Van der Waals" not "waal"

Correction done

Reviewer 3 Report

The authors have addressed many of the comments. However, it is my opinion that the rationale of the study, i.e. the description of the purpose of each of the analyses reported, in the framework of the underlying hypotheses, as well as the discussion of the results and their significance in proving the final thesis, are still missing and make the article not acceptable in its present form. In particular, these issues should be addressed both to introduce the description of the work carried out, and more fully in the conclusions to describe how the results obtained integrate with each other.

Author Response

Reviewer 3

1.       

The authors have addressed many of the comments. However, it is my opinion that the rationale of the study, i.e. the description of the purpose of each of the analyses reported, in the framework of the underlying hypotheses, as well as the discussion of the results and their significance in proving the final thesis, are still missing and make the article not acceptable in its present form. In particular, these issues should be addressed both to introduce the description of the work carried out, and more fully in the conclusions to describe how the results obtained integrate with each other.

The purpose of each analyses has been reported. The underlying hypotheses has been recast, as well as the discussion of the results.

In the conclusions, the results have been described showing how they integrate with each other

Round 3

Reviewer 1 Report

The manuscript by Iheagwam and colleagues underwent to a substantial improvement from the primary version with a better description of the methodological and results parts.

Nevertheless, there is an important notice to be point out regarding the molecular dynamic studies: there are no replicas of the three investigated protein-ligand systems. Normally, replicas of the same system are carried out to strength the statistical significance of the results. On the top of that, the authors also declare, in the ‘conclusion section’, to have performed docking studies using different docking algorithms and running different replicas of the dynamics, unfortunately this is not the case from the data provided in the manuscript.

Author Response

In the conclusion section, we did not declare to have performed docking studies using different docking algorithms and running different replicas of the dynamics. Rather, we noted them as limitations of the study.

Reviewer 3 Report

The authors globally improved the manuscript

Author Response

The authors appreciate the reviewers input